



# Impacts of emission changes in China from 2010 to 2017 on domestic and intercontinental air quality and health effect

Yuqiang Zhang[1], Drew Shindell[1,2,3], Karl Seltzer[1], Lu Shen[4], Jean-Francois Lamarque[5], Qiang Zhang[6], Bo Zheng[7], Jia Xing[8], Zhe Jiang[9], Lei Zhang[10]

[1]Nicholas School of the Environment, Duke University, Durham, NC, USA
[2]Duke Global Health Initiative, Duke University, Durham, NC, USA
[3]Porter School of the Environment and Earth Sciences, Tel Aviv University, Tel Aviv, Israel
[4]John A. Paulson School of Engineering and Applied Sciences, Harvard University, Cambridge, Massachusetts 02138, USA
[5]NCAR/UCAR, Boulder, CO 80301, USA
[6]Department of Earth System Science, Tsinghua University, Beijing 100084, China
[7]Institute of Environment and Ecology, Tsinghua Shenzhen International Graduate School, Tsinghua University, Shenzhen 518055, China
[8]State Key Joint Laboratory of Environmental Simulation and Pollution Control, School of Environment, Tsinghua University, Beijing 100084, China
[9]School of Earth and Space Sciences, University of Science and Technology of China, Hefei, Anhui, 230026, China
[10]State Key Laboratory of Pollution Control & Resource Reuse, School of the Environment, Nanjing University, Nanjing 210023, China

*Correspondence to*: Yuqiang Zhang (Yuqiang.Zhang@duke.edu)

**Abstract.** China has seen dramatic emission changes from 2010, especially after the implementation of Clean Air Action in 2013, with significant air quality and human health benefits observed. Air pollutants, such as $PM_{2.5}$ and surface ozone, as well as their precursors, have long enough lifetime in the troposphere which can be easily transported downwind. So emission changes in China will not only change the regional air quality domestically, but also affect the air quality in downwind regions. In this study, we use a global chemistry transport model to simulate the influence on both domestic and foreign air quality from the emission change from 2010 to 2017 in China. By applying the health impact functions derived from epidemiology studies, we then quantify the changes in air pollution-related (including both $PM_{2.5}$ and $O_3$) mortality burdens at regional and global scales. The majority of air pollutants in China reach their peak values around 2012 and 2013. Compared with the year 2010, the population-weighted annual $PM_{2.5}$ in China increases till 2011 (94.1 µg m$^{-3}$), and then begins to decrease. In 2017, the population-weighted annual $PM_{2.5}$ decreases by 17.6%, compared with the values in 2010 (84.7 µg m$^{-3}$). The estimated national $PM_{2.5}$ concentration changes in China are comparable with previous studies using fine-resolution regional models, though our model tends to overestimate $PM_{2.5}$ from 2013 to 2017 when evaluated with surface observation in China during the same periods. The emission changes in China increased the global $PM_{2.5}$-related mortality burdens from 2010 to 2013, by 27,700 (95%CI: 23,900—31, 400) deaths yr$^{-1}$ in 2011, and 13, 300 (11,400—15,100) deaths yr$^{-1}$ in 2013, among which at least 93% occurred in China. The sharp emission decreases after 2013 bring significant benefits for reduced avoided premature mortality in 2017, reaching 108, 800 (92,800—124,800) deaths yr$^{-1}$ globally, among which 92% happening in China. Different trend as $PM_{2.5}$, the annual maximum daily 8-hr ozone in China increased, and also the ozone-related premature deaths, ranging



from 3,600 (2,700—4,300) deaths yr⁻¹ in 2011 (75% of global total increased premature deaths), and 8,500 (6,500—9,900) deaths yr⁻¹ in 2017 (143% of the global total). Downwind regions, such as South Korea, Japan, and U.S. generally see a decreased O₃-related mortality burden after 2013 as a combination of increased export of ozone and decreased export of ozone precursors. In general, we conclude that the sharp emission reductions in China after 2013 bring benefits of improved air
quality and reduced premature deaths associated with air pollution at global scale. The benefits are dominated by the PM₂.₅ decreases since the ozone is shown to actually increase with the emission decrease.

## 1 Introduction

Fine particulate matter with an aerodynamic diameter of less than 2.5 μm (PM₂.₅) has been of particular interest to the research community as it has been known to pose great threats to environment, such as visibility impairment and material damages
(Hand et al., 2013, 2014; Wu and Zhang, 2018), and human health (e.g. Pope et al., 2002; Krewski et al., 2009). Increased relative risk were found between short-term PM₂.₅ exposures with daily all-cause mortality, respiratory and cardiopulmonary mortality across 24 countries in 652 cities (Liu et al., 2019). Long-term exposure to PM₂.₅ has been found to cause premature deaths from cardiopulmonary and respiratory disease (Krewski et al., 2009; Burnett et al., 2014). Short-term exposure to ozone were associated with hospital admissions, emergency room visits for respiratory causes and school absences (Katsouyanni et
al., 2009), and long-term exposure to surface ozone were also related with premature deaths from respiratory disease (Jerrett et al., 2009; Turner et al., 2016).

PM₂.₅, as well as its precursors, can travel long distances, affecting air quality and health in other receptor regions (Ewing et al. 2010; Pfister et al., 2011; Anenberg et al., 2014), despite its relatively short lifetime in the atmosphere (days to weeks). Tropospheric ozone has a much longer lifetime compared with PM₂.₅, with approximately a global average of 23 days (Young
et al., 2013), and the research community has expressed particular interest in studying its intercontinental transport (e.g. Zhang et al., 2008, 2014; Cooper et al., 2015; Lin M. et al., 2012, 2017; Parrish et al., 2014). Numerous studies have been carried out to investigate the source-reception relations on air quality and associated mortality burden from emission changes in one source region onto the others (West et al., 2009a,b; Fry et al., 2013; Crippa et al., 2019). The Task Force on Hemispheric Transport of Air Pollution (TF HTAP, Janssens-Maenhout et al., 2015), has made great effort to organize international scientists to study
the effect of emissions changes on the intercontinental transport of air quality and human health (http://www.htap.org/, accessed April 11 2020). Liang et al. (2018) used the ensemble model outputs from the TF HTAP, and estimated the source-receptor relations for air quality and avoided premature deaths from 20% reductions of anthropogenic emissions in East Asia. They estimated that 96,600 premature deaths would be avoided globally associated with PM₂.₅ reductions, with 6% (5, 500 deaths) in other regions, while for ozone, 11, 400 premature deaths would be avoided globally, with 15% (1,700 deaths) in
foreign countries other than East Asia.

To tackle the severe air pollution problem in China, the Chinese government has implemented strict clean air policies in recent years (State Council of the People's Republic of China, 2013). Before 2013, the clean air control policies were mainly focusing



on the emission standards of industry and power sectors, and after 2013 eight more stringent control measures were developed

after China committed to reducing $PM_{2.5}$ pollution (see Fig. 1 in Zheng B. et al., 2018b). Significant emission reductions have

been observed in China, and the air quality has substantially improved especially after the "Air Pollution Prevention and

Control Action Plan" (APPCAP) in 2013 (Zheng Y. et al., 2017; Zhang et al., 2019; UN Environment 2019). The relative

change of China's anthropogenic emissions for specific air pollutants during 2010–2017 are estimated as follows: −35 % for

primary $PM_{2.5}$, -62 % for $SO_2$, −27 % for BC, −35 % for OC, −17 % for $NO_x$, and −27 % for CO (Zheng B. et al., 2018b).

Significant decreases were observed for the $PM_{2.5}$ concentrations from surface observations, satellite retrievals and model

simulations (e.g. Song et al., 2017; Huang et al., 2018; Li Y. et al., 2015; Li C. et al., 2017; Lin C. et al., 2018; Zheng Y. et al.,

2017; Zhang Q. et al., 2019). The rapid reductions of major air pollutants in China were also confirmed using a long-term,

robust observational record at Fukue Island, Japan (Kanaya et al., 2020). A recent study using high-resolution regional air

quality model showed that the estimated national population–weighted annual mean $PM_{2.5}$ concentrations decreased from 61.8

to 42.0 µg m$^{-3}$ from 2013 to 2017 (Zhang Q. et al. 2019). Meanwhile, summertime daily maximum 8-h average ozone (MDA8)

in China has been shown an increasing trend since 2013 (Lu et al., 2018, 2020). The increasing surface ozone trend may be

partially explained by the slowing down of the aerosol sink of hydroperoxyl radicals which mainly reacts with NO to produce

ozone that caused by the significant $PM_{2.5}$ reductions, though a field campaign result may not be consistent with model

sensitivity results (Li et al., 2019; Tan et al., 2020). Recent studies have found that the major air pollutants, such as $SO_2$, $NO_x$,

CO, which are precursors for $PM_{2.5}$ and surface ozone, already reach peak values in 2012 or 2013, depending on the pollutants

(Zheng B. et al., 2018a,b).

Previous studies have evaluated the benefits from the APPCAP in China since 2013 on improved air quality including both

$PM_{2.5}$ and $O_3$ and avoided premature deaths (e.g. Huang et al., 2018; Zhang et al., 2019; Lu et al., 2020). However, limited

studies have been carried out to investigate the benefits of these actions on global air quality and air pollution related mortality

burden. In this study, we use a global chemical transport model to simulate the global air quality changes from 2010 to 2017

as a result of emission changes in China at the same time. Then we will apply the concentration response function (CRF) to

estimate the air pollution-related mortality burden changes in China and other countries. This particular time period was chosen

because the emissions in China were seen slightly increases and then significantly decreases. By incorporating the contrasting

emission trends in China, we want to compare effect of different emission trends in China on global air quality.

## 2 Data and Methods

### 2.1 Model simulation using the CAM-chem

The global air quality from 2010 to 2017 were simulated using the CAM-chem model (CAM version 4, Lamarque et al., 2012)

at a horizontal resolution of 1.9◦ (latitude) × 2.5◦ (longitude), and 56 vertical levels between the surface and 4 hPa (≈40 km),

driven by the NASA GEOS5 Global Atmosphere Forcing Data (Tilmes 2016, http://rda.ucar.edu/datasets/ds313.0/, last

Accessed 20 April, 2020). The detailed configurations, including the lower boundary conditions for long-lived species such as





CO$_2$ and CH$_4$, the online biogenic emission inventory, and other natural emissions, are referred to Zhang et al. (2016). In this version of CAM-chem, the bulk aerosol model was applied based on the work of Tie et al. (2001, 2005), in which the sulfate aerosol is formed by the oxidation of SO$_2$ in both the gas and aqueous phase (Lamarque et al., 2012; Tilmes et al., 2016). The ammonium nitrate is also represented depending on the amount of sulfate present in the air mass following the parameterization of gas/aerosol partitioning by Metzger et al. (2002). A comprehensive evaluation for the model performance in simulating

temporal and spatial distribution of global ozone and aerosols by comparing surface observation, balloon, aircraft and satellite were carried out in previous studies (Tilmes et al., 2015, 2016; Zhang et al., 2016). In this study, we used the same configurations as our previous one (Zhang et al., 2016), and in the Chemistry-Climate Model Initiative project (Tilmes et al., 2016). We focus on the model evaluation in simulating the surface ozone and PM$_{2.5}$ concentrations in China from 2013 to 2017. Base case simulations were run consistently from 2010 to 2017 with one year spin-up in 2009 using time-varying global

anthropogenic emission inventory as a combination of the Community Emissions Data System (CEDS, v2017-05-18, Hoesly et al., 2018) developed by the Pacific Northwest National Laboratory, and the Multi-resolution Emission Inventory (MEIC, http://www.meicmodel.org/, last access 20 April 2020) developed by Tsinghua University, China. We performed first sensitivity simulation (CEDS_MEIC_ChinaFix) which keeps China emissions constant at the level of 2010, and the differences between the base and CEDS_MEIC_ChinaFix are the influences of anthropogenic emission changes in China since 2010 on

global air quality and human health (Table 1). By making comparisons between these two scenarios, we can also rule out the influences of meteorological variabilities on the global distribution of ozone and PM$_{2.5}$. The global anthropogenic emissions other than China after 2014 were kept constant, since no global emissions inventory is available after 2014 when we first prepared the study. McDuffie et al. (2020) updated the global CEDS anthropogenic emissions through 2017 with continued update into 2019 (https://github.com/JGCRI/CEDS, last access 6 May 2021). However, this would have negligible effect on

our conclusions since our focus is the emission changes from China on influence of domestic and international air transport. We also performed another sensitivity simulation with global anthropogenic emissions keeping constant at 2010 level (CEDS_MEIC_GlobalFix, Table 1). The air quality changes from 2011 to 2017 relative to 2010 from this sensitivity could allow us to take a look at the meteorological changes on PM$_{2.5}$ and O$_3$ changes in China at the same time.

It also came to our attention that the CEDS emissions tends to overestimate the magnitude of the Chinese emissions, and

underestimate the emission decreasing trend in China (Zheng B. et al., 2018b; Paulot et al., 2018). From Fig. S1, we see that in 2014, the emissions from CEDS are at least 20% higher than the estimations from MEIC for most of the air pollutants, except for non-methane volatile organic compounds (1%). More specifically, the SO$_2$, OC and BC emissions estimated in CEDS are 84%, 81% and 58% higher individually than those estimated in MEIC. For the emission trend, CEDS estimated a continued increasing trend while MEIC estimated a peak for most of the air pollutants before the year 2012 (Liu et al., 2016;

Zheng B. et al., 2018b). From Fig. S2 which shows the spatial patterns of the emission differences between CEDS and MEIC in 2014, we can see that the emissions in CEDS are higher in the western and south China, and lower in the eastern China. We performed another sensitivity simulation which applied CEDS global anthropogenic only from 2010 to 2014 (CEDS_Global),



to evaluate the model's performance in simulating $PM_{2.5}$ and $O_3$ in China and also discuss the relative air quality changes applying different emission inventory in China.

## 2.2 Surface observation for $PM_{2.5}$ and $O_3$ in China

Surface observation for hourly $PM_{2.5}$ and $O_3$ concentration were downloaded from China National Environmental Monitoring Center (CNEMC) Network (http://106.37.208.233:20035/) from 2013 to 2017, since data was not available before 2013. We evaluated the model's performance in simulating annual $PM_{2.5}$ and maximum daily 8 h average (MDA8 $O_3$) as these two metrics are related with the health impact analysis we performed in our study.

## 2.3 Health impact assessment for surface $PM_{2.5}$ and $O_3$

We applied the health impact function derived from long-term cohort studies, together with the baseline mortality rates and exposure population to quantify the air quality related mortality burden changes. The mortality burdens related to ambient air pollution including $PM_{2.5}$ and $O_3$ are calculated following Eq. (1):

$$\Delta Mort = Y_0 \times AF \times Pop \ , \qquad\qquad (1)$$

Where $\Delta Mort$ is mortality burden attributed to surface $PM_{2.5}$ and $O_3$, $Y_0$ is the baseline mortality rates for cause of specific disease, $AF$ is the attribution fraction calculated as $1 - \frac{1}{RR}$ with $RR$ as the relative risk, and $Pop$ is the exposed population with ages greater than 25 years old. The $RR$ for the $PM_{2.5}$ is calculated using the latest integrated exposure response model (IER, Burnett et al., 2014), following the previous methods in our group (Shindell et al., 2018). We used the 1000 simulations for the parameter distributions of α, β and γ retrieved from the latest GBD study (Stanaway et al., 2018) to derive the mean $RR$ with 95% uncertainty intervals. The $RR$ for the long-term exposure to surface ozone is from the updated cohort study (Turner et al., 2016), with $RR$ of 1.12 (95 % confidence interval (CI): 1.08, 1.16) for respiratory disease. Country-age-specific baseline mortality rates ($Y_0$) in 2010 were retrieved from the latest GBD (Global Burden of Disease) project (Stanaway et al., 2018), and remapped to match the 10[th] International Statistical Classification of Diseases and Related Health Problems codes as used in the cohort study (Turner et al., 2016; Seltzer et al., 2020).

## 3 Results

### 3.1 Model evaluation with surface observation in China

From the model evaluation metrics, we see that the CAM-chem base simulation (CEDS_MEIC scenario, see Table 1) generally overestimates the annual $PM_{2.5}$ concentration in China, with mean bias (MB) of 19.3 µg m[-3] and normalized mean bias (NMB) of 37.2% for all the 5 years. The MBs are around 20 µg m[-3] from 2014 to 2017 with lowest values in year 2013 (MB of 7.6 µg m[-3]) and highest in 2015 (21.7 µg m[-3]). The lower MB and NMB in 2013 could be caused by the much less data available in



2013. The positive NMBs for all the years shows that the overestimations are systematic and may not affect our main conclusions since we focus on the changes among years. The higher modeling bias for surface $PM_{2.5}$ from the CAM-chem were also seen in other studies, for example, He and Zhang (2014) reported NMB of 37.6% and 41.85% for the contiguous United States and Europe in 2001. The bias in simulating the surface $PM_{2.5}$ in the CAM-chem (version 4) was mainly caused

by the inaccurate prediction of $SO_4^{2-}$, $NH_4^+$, and organic aerosols, and missing major inorganic aerosol species such as nitrate and chloride (He and Zhang, 2014; Tilmes et al., 2016). By including advanced inorganic aerosol treatments, such as condensation of volatile species, explicit inorganic aerosol thermodynamics for sulfate, ammonium, nitrate, sodium, and chloride (He and Zhang, 2014), and more comprehensive secondary organic aerosols approach (Volatility Basis Set scheme, Times et al., 2019; Liu et al., 2020), the performance for simulating surface $PM_{2.5}$ could be significantly improved. The

different metrics also exhibits small annual variabilities, with NMB around 40% and NME around 50% for each year, except for 2013 and 2014 which have smaller NMB values due to the limited size of the observations (Table 2). The CAM-chem can generally reproduce the spatial patterns of the annual $PM_{2.5}$ distributions, with correlation coefficient (R) greater than 0.7. With CEDS emissions applying globally (CEDS_Global scenario, see Table 1), we have better performance in both 2013 and 2014 for the annual $PM_{2.5}$ evaluation. Part of the reason is that we have less available data in these two years, while we suspect the

main reason is that though total emissions in China are higher in CEDS than those in MEIC, CEDS tends to underestimate the emissions in eastern and central China (Fig. S2), where the majority of observations are available. With CEDSs emission applied, the annual $PM_{2.5}$ is lower in eastern China and higher in western and northwestern China (Fig. S3). The performance for CAM-chem in simulating annual MDA8 $O_3$ is slighter better than the surface $PM_{2.5}$, with NMB lower than 20% for all the years (Table 3). From Table 3, we can also see that the CAM-chem overestimates the annual MDA8 $O_3$ in China, which means

our estimation for the $O_3$-related mortality burden will have positive bias. A high bias of about 10 ppb can be attributed to the coarse model resolution, which leads to an overestimate of ozone production, because of diluted emissions of ozone precursors (Tilmes et al, 2015). For annual MDA8 $O_3$, the CEDS simulation (CEDS_Global scenario) has a poorer performance compared with using MEIC emissions, which maybe caused by the overestimation of the surface ozone in western China (Fig. S4).

**3.2 Air quality changes in China from 2010 to 2017**

We first report the trends for the national annual population-weighted (Pop-weighted) $PM_{2.5}$ which decreased by by 17.6%, changing from 84.7 µg m$^{-3}$ in 2010 to 69.8 µg m$^{-3}$ in 2017 (Fig. 1). The Pop-weighted $PM_{2.5}$ was highest in 2011 (annual average of 94.1 µg m$^{-3}$), and decreases until 2017 (69.8 µg m$^{-3}$). From 2013 to 2017, the national annual Pop-weighted $PM_{2.5}$ decreased by 15.9 µg m$^{-3}$, comparable to the values (19.8 µg m$^{-3}$) estimated by Zhang Q. et al. (2019) which used a high-resolution regional air quality model. The area-weighted national average $PM_{2.5}$ concentrations shares a similar trend as the

Pop-weighted average but with much lower values (Fig. 1a), demonstrating the fact that the high $PM_{2.5}$ pollutions happen in more density region. The annual average of area-weighted $PM_{2.5}$ concentration decreased by 7.6 µg m$^{-3}$, consistent with the estimations by Ding et al. (2019a) at 9.0 µg m$^{-3}$, which also demonstrates that overestimations of annual $PM_{2.5}$ from CAM-chem are systematic and will not affect our trend discussions. For the spatial patterns of the $PM_{2.5}$ changes, we see that



significant annual $PM_{2.5}$ changes (increases before 2013 and decreases after then) mainly occur in the eastern China (Fig. 2),
which were the focused regions for China APPCAP (Ding et al., 2019a,b). When distinguishing the contributions from
anthropogenic emissions vs. meteorology, we find that the annual $PM_{2.5}$ decreases in China are mainly dominated by the
emission changes, consistent with previous studies (Dang and Liao, 2019; Ding et al., 2019a; Zhai et al., 2019; Zhang Q. et
al., 2019). Compared with the year 2010, interannual meteorology led to annual $PM_{2.5}$ decreases by as high as 3.5 µg m$^{-3}$ in
2012, and increases as high as 3.1 µg m$^{-3}$ in 2015 (Fig. 4a).

Different from the $PM_{2.5}$ trend, the annual Pop-weighted average MDA8 $O_3$ has a continued increasing trend since 2010 (Fig.
1b), with peaks in 2014 (59.5 ppbv), and then decreases to 2017 (57.1 ppbv). The area-weighted MDA8 $O_3$ was comparable
or even larger than the Pop-weighted (e.g. 2012), as a result of more uniform $O_3$ distribution in China or even higher ozone
events in western China because of stratosphere-troposphere exchange with less population (Wang et al., 2011; Li et al., 2019).
For the spatial patterns, the ozone increases mainly in Beijing–Tianjin–Hebei and Yangtze-River-Delta, and slightly decreases
in the south (Fig. 3; Fig. S5). The anthropogenic emission reductions in China leads to ozone increases (Fig. 4b), which could
be partially explained by aerosol sink of hydroperoxyl radicals slowing down due to $PM_{2.5}$ decreases (Li et al., 2018). The
interannual meteorological condition changes have a much larger positive effect on the annual MDA8 $O_3$, compared with the
anthropogenic emission changes, leading to ozone increases as high as 8.1 ppbv in 2014 and as low as 0.7 ppbv in year 2011.
The meteorology-induced ozone increases can be attributed to increasing temperature which enhances the ozone production
and biogenic NMVOCs emissions (Ding et al., 2019b; Liu and Wang, 2020), and the increases solar radiation (Wang et al.,
2020; Ma et al., 2021).

### 3.3 Emission changes in China on global air quality and health

### 3.3.1 Global and regional air quality

The simulated global tropospheric ozone burden (total ozone below the chemical tropopause of 150 ppbv) from 2010 to 2017
calculated from the CEDS_MEIC simulation is 327.5 ± 5.2 Tg, agreeing well with the present tropospheric ozone burden
estimated from previous ensemble models (ACCENT: 336 ± 27 Tg; ACCMIP:337 ± 23 Tg; TOAR: 340 ± 34 Tg, and CMIP6:
348 ± 15 Tg; Griffiths et al., 2021). From Fig. 5, the change for global tropospheric ozone burden from the emission changes
in China ranges from 0.6 Tg (2011 and 2012) to -1.9 Tg (2017). The tropospheric ozone burden changes are not only seen in
China, but also in downwind regions, especially in Northern Hemisphere, such as Pacific Ocean and U.S., as a result of the
vertical transport of the air pollutants (Fig. S6).

Due to the prevailing western wind, the air pollutants in China could be easily transported to downwind regions, especially
during springtime (Lin M. et al., 2012; Liang et al., 2019). From Fig. 6, we see that downwind regions, South Korea has the
largest pop-weighted $PM_{2.5}$ changes, ranging from 0.7 µg m$^{-3}$ in 2012 to -2.63 µg m$^{-3}$ in 2017, following by Japan and U.S.
(Fig. 6a). However, for ozone, we see that the emission changes in China have increased the surface ozone in South Korea
from 2010 to 2017, mainly contributed by the increased export of ozone. Both Japan and U.S. are firstly seen an increase and





then decrease. However, the increases and the decreases are not usually following the same year as emissions and PM$_{2.5}$ concentration changes, demonstrating the non-linearity of the ozone productions.

### 3.3.1 Global and regional air pollution-related mortality burden changes

The global ambient PM$_{2.5}$-related mortality burden in 2010 is 4.08 million (95%CI: 2.15—6.0 million), similar with previous

estimated in the same year applying the same IER method (3.6±1.0 million in 2010, Shindell et al., 2018). Compared with 2010, the emission changes in China (by comparing CEDS_MEIC and CEDS_MEIC_ChinaFix) lead to mortality increases by 27, 700 deaths yr$^{-1}$ (95%CI: 23, 900—31, 400 deaths yr$^{-1}$) in 2011, with 93% occurring in China (25, 800, 95%CI: 22, 300—29, 200 deaths yr$^{-1}$; Table 4). From Table 4 (last column), we can see that China takes the largest majority of global PM$_{2.5}$-mortality burden changes from 2011 to 2017 (more than 90%), because of the relative linear relations between emission

and concentration for PM$_{2.5}$, and its relative shorter lifetime in the troposphere. Relative to year 2010, the emission changes in China in 2017 leads to 108, 800 (95%CI: 92,800—124, 800) avoided premature deaths, with 92% (95%CI: 85, 900—114, 300) occurring in China. Among the three downwind regions, Japan displays the largest influences for the PM$_{2.5}$-related mortality burden changes, ranging from 197 deaths yr$^{-1}$ in 2011 to -875 deaths yr$^{-1}$ in 2017. The emission changes in China have comparable effect on the PM$_{2.5}$-related mortality burden change among South Korea and U.S. (Table 4).

The global ozone-related mortality in 2010 is 1.02 million (95% CI: 0.73—1.28 million), consistent with previous estimations using other global CTMs, such as GEOS-Chem (1.04—1.23 million) and GISS (0.8—1.3 million), applying the same relative risk (Mally et al., 2017; Shindell et al., 2018). In 2011, the emission changes in China increased the global ozone-related mortality by 4,900 (95%CI, 3,700—5,900) deaths yr$^{-1}$ (Table 5), among which 73% is happening in China (3600 deaths yr$^{-1}$, 95%CI: 2,700—4,300). For the three downwind region, South Korea, Japan, and U.S., the added O$_3$-related mortality burdens

in each country are 23, 200 and 131 deaths yr$^{-1}$ individually. In 2017, the reduced anthropogenic emissions in China greatly increased the ozone-related mortality burden, by 8,500 deaths yr$^{-1}$, which is 43% higher than the global total added O$_3$-related mortality burden (5,920 deaths yr$^{-1}$). In general, the downwind regions have decreased O$_3$-related mortality burden (-65 and -289 deaths yr$^{-1}$ for Japan and U.S. individually, Table 5), except for South Korea, which also has increased mortality burden by 17 deaths yr$^{-1}$ (Table 5).

## 4 Discussion

Dramatic changes are observed for anthropogenic emissions in China since 2010, with majority of air pollutants, such as NO$_x$, SO$_2$, are CO peaks around 2012 and 2013, and decreases afterwards significantly. In this study, we use a global chemistry transport model (CAM-chem) to simulate the emission changes in China on domestic and global air quality and air pollution-related mortality burden changes from 2010 to 2017. Compared with surface PM$_{2.5}$ observations network in China, our model

tends to overestimate the annual PM$_{2.5}$ concentrations, with normalized mean bias (NMB) of 37.2% and normalized mean error (NME) of 52.0% from 2013 to 2017. We also evaluated the model performances by applying the regional emission





inventory (MEIC) developed by Tsinghua University and the global emission inventory (CEDS) developed by PNNL and applied extensively in the CMIP6 experiments. For surface $PM_{2.5}$, we find that the CAM-chem with CEDS emissions tends to have a lower NMB and NME in 2013-2014 since CEDS emissions are lower in urban areas in China than that from MEIC though the national totals from CEDS are much higher than those from MEIC. For surface $O_3$, CAM-chem with MEIC emissions in China has a lower NMB (13.7%) and NME (21.9%) for the annual MDA8 $O_3$. The simulations with CEDS in China tend to have larger NMB (15.2%) and NME (36.6%) in 2013-2014.

From 2010 to 2017, we calculate that, the annual average population-weighted (Pop-weighted) $PM_{2.5}$ increases in China till 2011 (94.1 µg m$^{-3}$), and then decreases sharply afterwards. The annual Pop-weighted $PM_{2.5}$ in 2017 decreases by 17.6% (-14.9 µg m$^{-3}$), compared with the value in 2010 (84.7 µg m$^{-3}$), and 18.5% lower than 2013 (85.8 µg m$^{-3}$). Though CAM-chem overestimates the $PM_{2.5}$ concentrations in China, the simulated decreasing trend for annual $PM_{2.5}$ from 2013 to 2017 (-15.9 µg m$^{-3}$ for Pop-weighted, and 7.6 µg m$^{-3}$ for area-weighted) is comparable with previous studies using high resolution regional air quality models (-19.8 µg m$^{-3}$ estimated from Zhang Q. et al. (2019), and -9.0 µg m$^{-3}$ from Ding et al. (2019a)). The overestimation of the surface $PM_{2.5}$ concentration in China from CAM-chem were unlikely to affect our estimations of the trends for the $PM_{2.5}$-related health benefits, because of the high $PM_{2.5}$ concentration in China, as well as the non-linearity of the IER functions with the $PM_{2.5}$ concentration (Zhang Q. et al., 2019). The $PM_{2.5}$ overestimations in the CAM-chem model were likely caused by the uncertainties in the bottom-up emission inventories (Shen et al., 2019; Zhang Q. et al., 2019), and missing chemical mechanisms for the $PM_{2.5}$ components (Tilmes et al., 2016; Liu et al., 2020), and are in the same magnitudes as seen in other high resolution regional models (25%-30% in Shen et al., 2019; ~20% in Zhang Q. et al., 2019). The emission changes in China from 2010 to 2013 increased the global $PM_{2.5}$-related mortality burdens, varying from 27, 700 deaths yr$^{-1}$ (95% confidence interval (CI): 23,900—31, 400) in 2011, and 13, 300 (95%CI: 11,420—15, 110) deaths yr$^{-1}$ in 2013. Among the increased premature deaths, at least 93% occurring in China. The sharp emission decreases after 2013 bring significant benefits for reduced avoided premature mortality in 2017, reaching 108, 800 (92,800—124, 800) deaths yr$^{-1}$, among which 92% (100, 100 deaths yr$^{-1}$ with 95%CI: 85, 900—114, 300) happening in China. Downwind regions, such as South Korea, Japan, and U.S. share the same $PM_{2.5}$ trend as China. The transport of $PM_{2.5}$ and its precursors could change the annual Pop-weighted $PM_{2.5}$ ranges from 0.7 µg m$^{-3}$ in 2011 and -2.6 µg m$^{-3}$ in 2017, leading to added 98 premature deaths in 2011, and 386 avoided premature deaths in 2017. Japan has a smaller change for the annual Pop-weighted $PM_{2.5}$, but much larger changes in $PM_{2.5}$-related mortality burden changes, ranging from 197 added premature deaths in 2011, and 875 avoided premature deaths in 2017. The influence for U.S. ranges from 44 added premature deaths in 2011, and 381 avoided premature deaths in 2017. Different trend as $PM_{2.5}$, the emission changes in China had an increasing trend for the annual maximum daily 8-hr ozone in China, and also the ozone-related premature deaths, ranging from 3, 600 deaths yr$^{-1}$ in 2011 (75% of global total increased premature deaths), and 8, 500 deaths yr$^{-1}$ in 2017 (143% of global total). Downwind regions, such as South Korea, Japan, and U.S. are generally seen decreased $O_3$-related mortality burden after 2013 as a combination of increased export of ozone and decreased export of ozone precursors. In general, we conclude that the sharp emission reductions in China after



2013 bring benefits of improved air quality and reduced premature deaths associated with air pollution at the global scale. The benefits are dominated by the $PM_{2.5}$ decreases.

**Data availability.** Global anthropogenic emissions data from CEDS are available from https://www.geosci-model-dev.net/11/369/2018/ (accessed May 4[th], 2020). MEIC emission inventory is available from http://meicmodel.org/?page_id=560 (last access May 6[th], 2021). Baseline health and population data are available from the World Health Organization and the United Nations, respectively. The CAM-chem model is available at http://www.cesm.ucar.edu/models/cesm1.2/ (accessed May 4[th], 2020). Data from CESM modelling that support the findings of this study are available from the corresponding author upon request.

**Author contributions.** YZ and DS originally designed the study, and YZ conducted all simulations, created all figures and wrote the manuscript, with comments and edits from all authors. BZ and QZ contributed to develop the MEIC emission inventory.

**Competing interests**. The authors declare that they have no conflict of interest.

**Acknowledgements**. Y.Z. and D.S. acknowledge the support by the NASA GISS grant 80NSSC19M0138. We gratefully acknowledge the CESM model which was developed and distributed by NCAR. We also appreciate the efforts from China Ministry of Ecology and Environment for maintaining the nationwide observation network and publishing hourly $PM_{2.5}$ and $O_3$ concentrations. We would like to thank the University of North Carolina at Chapel Hill and the Research Computing group for providing computational resources and support that have contributed to these research results.

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





**Table 1: Model simulation performed for this study.**

| Name | Anthropogenic Emissions | Meteorology |
|---|---|---|
| CEDS_MEIC | MEIC in China from 2010 to 2017 <br> CEDS outside China from 2010 to 2017[1] | 2010 to 2017 |
| CEDS_MEIC_ChinaFix | MEIC in China constant as in 2010 <br> CEDS outside China from 2010 to 2017[1] | 2010 to 2017 |
| CEDS_MEIC_GlobalFix | Emissions kept constant at 2010 level from both MEIC and CEDS | 2010 to 2017 |
| CEDS_Global | 2010-2014 from CEDS globally | 2010 to 2014 |

[1]The global emissions other than in China after 2014 are the same as in 2014 values since it is not available.



**Table 2: Model performance for the annual PM$_{2.5}$ concentration compared with surface observation in China from 2013 to 2017, with mean bias (MB, µg m$^{-3}$), normalized mean bias (NMB, %), normalized mean error (NME, %), and root-mean-square error (RMSE, µg m$^{-3}$).**

| Year | # of stations | MB (µg m$^{-3}$) | NMB (%) | NME (%) | RMSE (µg m$^{-3}$) |
|---|---|---|---|---|---|
| 2013 | 378 | 6.9 | 9.5 | 29.4 | 27.3 |
| 2014 | 537 | 21.8 | 34.3 | 45.6 | 39.7 |
| 2015 | 1438 | 21.6 | 41.3 | 54.0 | 39.5 |
| 2016 | 1431 | 19.2 | 40.0 | 54.5 | 36.5 |
| 2017 | 1461 | 19.4 | 42.2 | 55.7 | 34.8 |
| 2013-2017 | 5245 | 19.3 | 37.2 | 52.0 | 26.7 |
| CAM_Chem using CEDS emissions only[1] | | | | | |
| 2013[1] | 378 | 0.8 | 1.0 | 23.5 | 20.1 |
| 2014[1] | 537 | 19.5 | 30.1 | 42.1 | 34.3 |
| 2013-2014[1] | 915 | 11.8 | 17.5 | 33.9 | 29.5 |

[1]The CAM-chem simulations applying global CEDS emissions only, which only has been ran from 2010 to 2014.




**Table 3: As Table 2 but for annual MDA8 O₃, with mean bias (MB, ppbv), normalized mean bias (NMB, %), normalized mean error (NME, %), and root-mean-square error (RMSE, ppbv).**

| Year | # of stations | MB (ppbv) | NMB (%) | NME (%) | RMSE (ppbv) |
|---|---|---|---|---|---|
| 2013 | 1029 | 4.8 | 11.6 | 20.2 | 10.6 |
| 2014 | 1033 | 7.2 | 17.5 | 24.1 | 12.4 |
| 2015 | 1026 | 6.0 | 14.4 | 23.0 | 11.9 |
| 2016 | 1031 | 6.5 | 15.7 | 22.7 | 11.8 |
| 2017 | 1042 | 3.8 | 9.3 | 19.5 | 10.3 |
| 2013-2017 | 5161 | 5.7 | 13.7 | 21.9 | 11.4 |
| CAM_Chem using CEDS emissions only[1] | | | | | |
| 2013[1] | 1029 | 14.1 | 33.8 | 34.5 | 16.4 |
| 2014[1] | 1033 | 16.3 | 39.2 | 39.6 | 18.3 |
| 2013-2014[1] | 2062 | 15.2 | 36.6 | 37.0 | 17.4 |

[1] The CAM-chem simulations applying global CEDS emissions only, which only has been ran from 2010 to 2014.



**Table 4. The changes for the PM₂.₅- mortality burden under the emission changes in China from 2010 to 2017 in China, as well as three other downwind regions—South Korea, Japan and U.S. The mortality burden changes at global level are also included. Positive values mean emission change in China increase the PM₂.₅-related mortality burden in this region, and negative values**
**mean decreases the PM₂.₅-related mortality burden.**

| Year | China | South Korea | Japan | U.S. | Global | % (China/Global) |
|---|---|---|---|---|---|---|
| 2011 | 25,800 | 98 | 197 | 44 | 27,700 | 93% |
| 2012 | 26,200 | 80 | 143 | 24 | 27,900 | 94% |
| 2013 | 12,600 | -21 | -122 | -39 | 13,300 | 95% |
| 2014 | -19,620 | -147 | -306 | -138 | -21,800 | 90% |
| 2015 | -47,670 | -226 | -541 | -191 | -51,600 | 92% |
| 2016 | -77,600 | -264 | -615 | -284 | -83,200 | 93% |
| 2017 | -100,100 | -386 | -875 | -381 | -108,800 | 92% |





**Table 5. As Table 4 but for ozone-related mortality burden changes.**

| Year | China | South Korea | Japan | U.S. | Global | % (China/Global) |
|---|---|---|---|---|---|---|
| 2011 | 3,600 | 23 | 172 | 131 | 4,900 | 73% |
| 2012 | 3,400 | 15 | 113 | 140 | 4,900 | 70% |
| 2013 | 5,500 | 17 | 115 | 93 | 6,600 | 83% |
| 2014 | 6,400 | 25 | 68 | -5 | 6,500 | 99% |
| 2015 | 7,400 | 22 | 34 | -100 | 6,500 | 113% |
| 2016 | 7,500 | 11 | -56 | -188 | 5,600 | 133% |
| 2017 | 8,500 | 17 | -65 | -289 | 5,900 | 143% |




**Figure 1: National population-weighted (Pop-Weighted) and area-weighted annual PM₂.₅ (a), and MDA8 O₃ (b) from 2010 to 2017 from our base model simulation (CEDS_MEIC). The units are μg m⁻³ for PM₂.₅ and ppbv for ozone.**

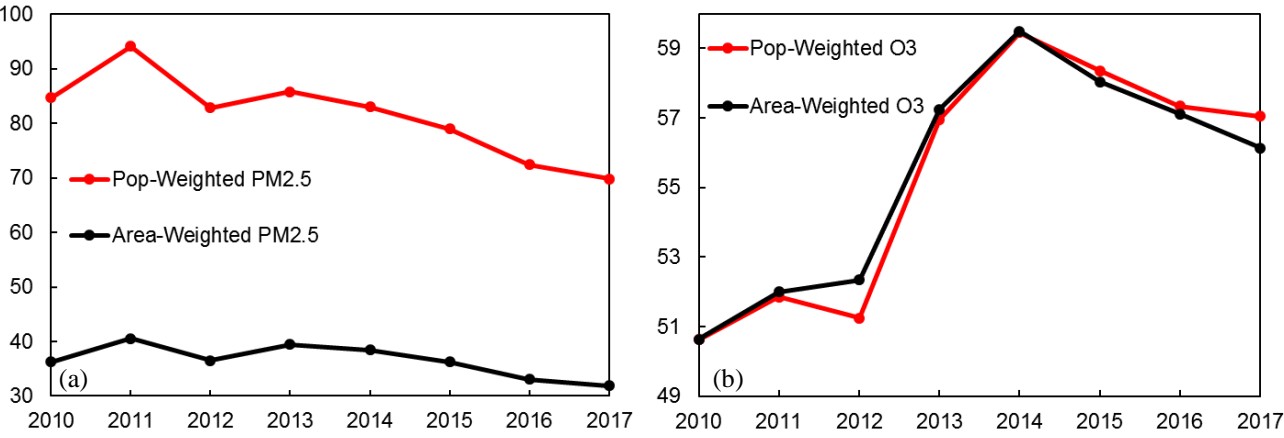

**Figure 2: Annual PM₂.₅ changes (unit of μg m⁻³) from 2011 to 2017 due to anthropogenic emission changes in China only. The results are calculated as the differences between CEDS_MEIC and CEDS_MEIC_ChinaFix for each year.**

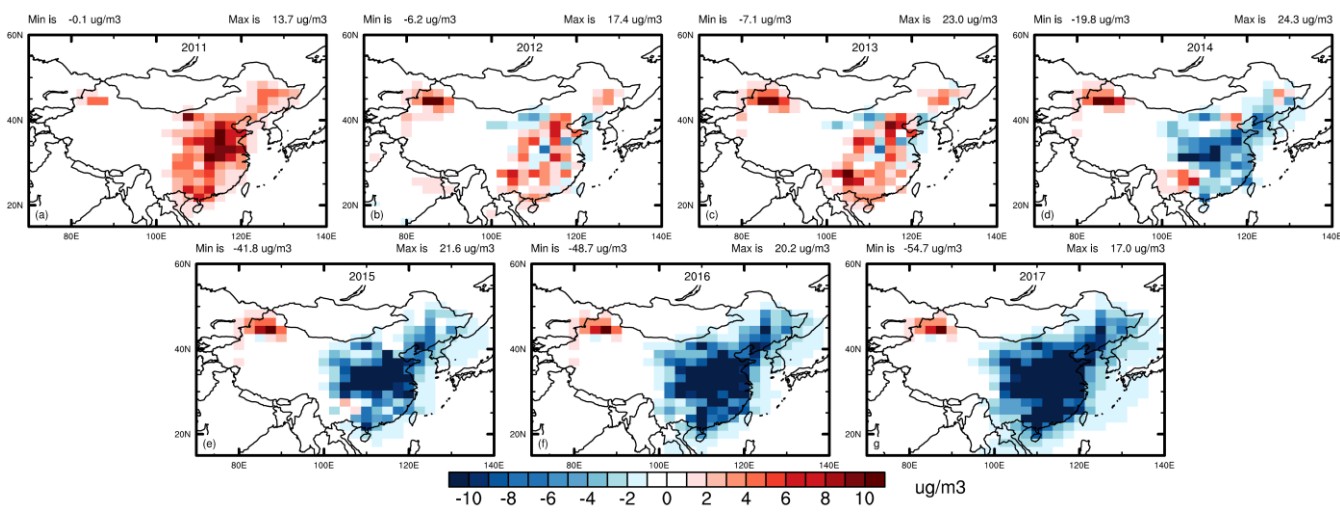





**Figure 3: Same as Fig. 2 but for annual MDA8 O₃ changes.**

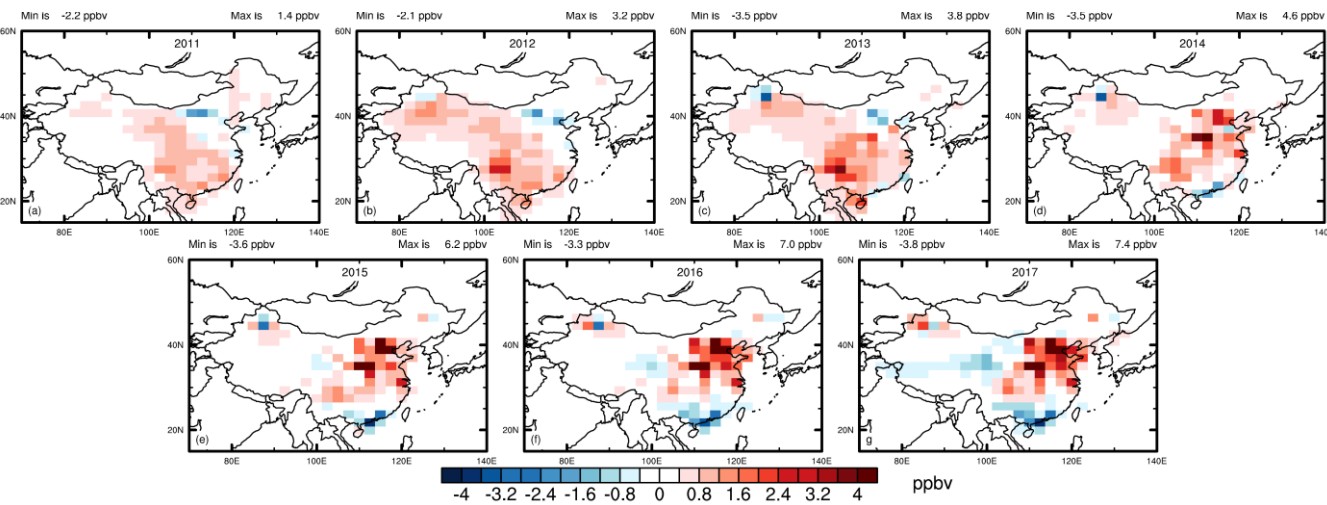



**Figure 4: The annual PM₂.₅ and MDA8 O₃ changes in China due to emissions and meteorological changes from 2010 to 2017.**

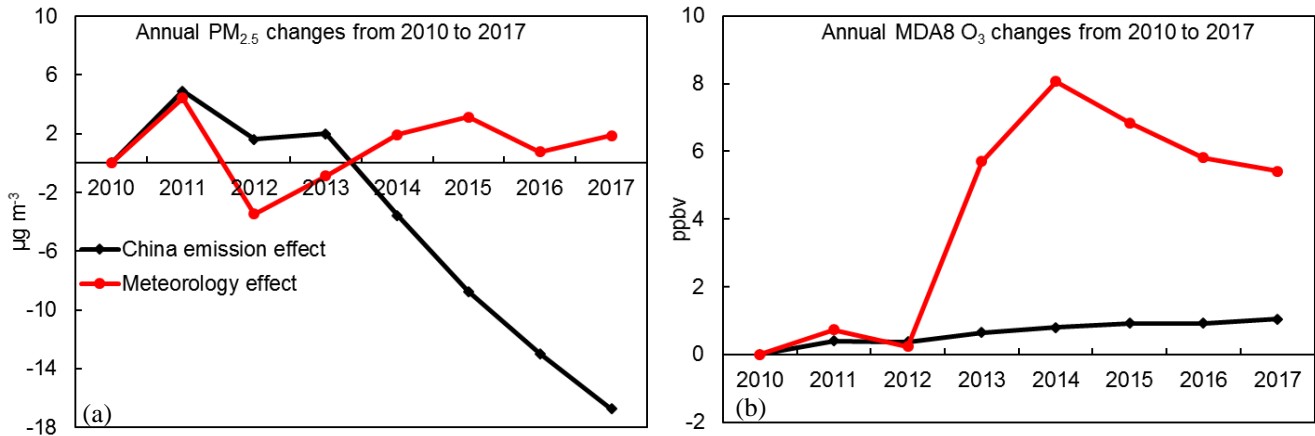






**Figure 5. Spatial distribution for global tropospheric ozone burden changes from 2010 to 2017, as a result of emission changes in China.**

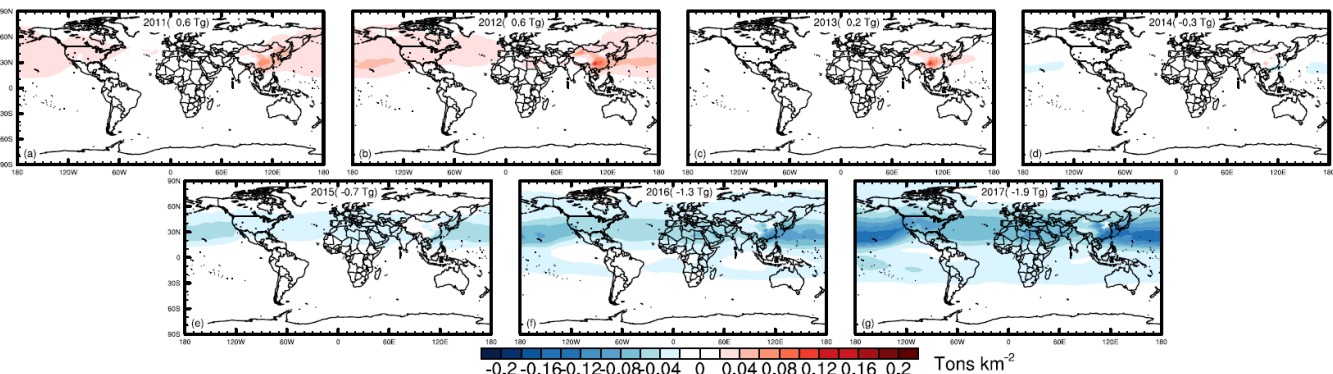

**Figure 6: The annual population-weighted PM2.5 and MDA8 O3 changes in South Korea, Japan, and U.S. from 2010 to 2017 on caused by Chinese emission changes.**

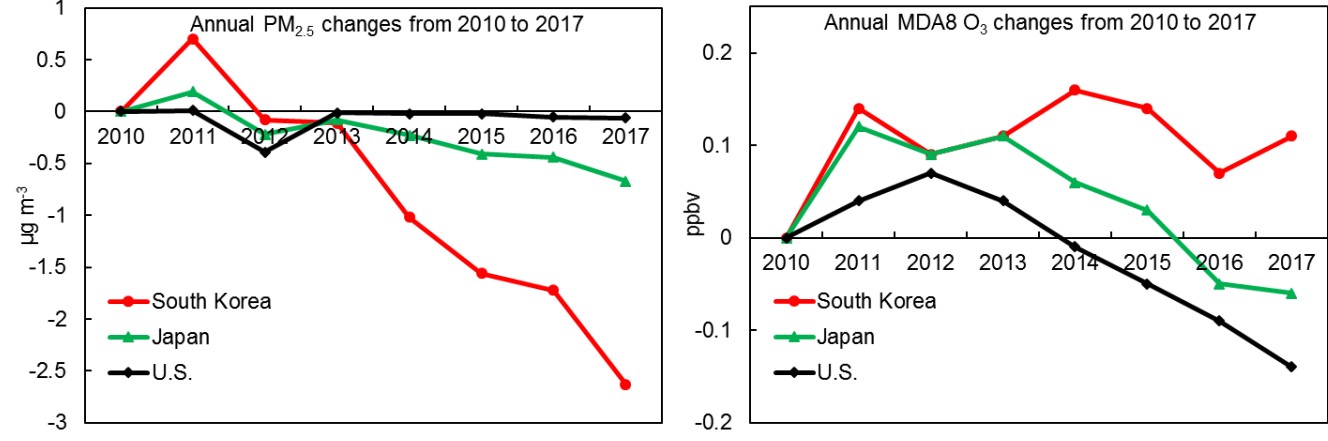