# Peer review of "Impacts of emission changes in China from 2010 to 2017 on domestic and intercontinental air quality and health effect"

_Atmospheric Chemistry and Physics, 2021_

## Author Comment (AC1)

**Response to comments #1**

Response: We thank the reviewer's helpful and constructive comments. We have made several modifications and implemented the suggestions as needed. We describe a few major changes, followed by our response to individual comments.

RC1 comments:

The manuscript by Zhang and colleagues models the change in ozone and $PM_{2.5}$ and related health impacts in China and downwind countries due to emission changes in China after the implementation of APPCAP. Overall, most of the results presented in this paper on decrease in $PM_{2.5}$ in China post APPCAP is not new and has been established in multiple studies involving models/measurements. The new aspect is assessing its impact in details on $PM_{2.5}$ and ozone exposure and related mortality burden in downwind regions, though the TF-HTAP initiative partly addresses this.

Response: We appreciate the reviewer's efforts in providing constructive comments for our paper. Though we made several significant changes based on the reviewer's comments below, we disagree with the reviewer's opinion that our study was not innovative enough. First, our study's time frame covers 2010 to 2017, which is before and after implementation of APPCAP in China in 2013, unlike the other studies (e.g., Zheng Y. et al., 2017; Huang et al., 2018; Zhang et al., 2019; Ding et al., 2019a, b; Li et al., 2019 and so on) which only emphasized the air quality and health improvements after implementation of APPCAP in 2013. Meanwhile, in our study, we explored how the emission changes in China, including both emission increases and decreases, have influences on both domestic and downwind regions' air quality and human health, while the other studies mentioned above only discussed the air quality changes in China. Second, the studies in the TF-HTAP (Liang et al., 2018, as pointed out by the reviewer) investigated the changes in air quality and human health from hypothetical 20 % anthropogenic emission reductions from East Asia (including China, Japan, Korea), while we used the up-to-date high-resolution bottom-up emission developed by our coauthors in China from 2010 to 2017, with emission changes of $-62\,\%$ for $SO_2$, $-17\,\%$ for $NO_x$, $+11\,\%$ for nonmethane volatile organic compounds (NMVOCs), $+1\,\%$ for $NH_3$, $-27\,\%$ for CO, $-35\,\%$ for $PM_{2.5}$, $-27\,\%$ for BC and $-35\,\%$ for OC (Zheng et al., 2018). In conclusion, our study is innovated in studying both the domestic and downwind regions' air quality changes from emission changes in China from 2010 to 2017, which was also acknowledged by reviewer 2.

References:

Ding, D., Xing, J., Wang, S., Liu, K. and Hao, J.: Estimated Contributions of Emissions Controls, Meteorological Factors, Population Growth, and Changes in Baseline Mortality to Reductions in Ambient $PM_{2.5}$ and $PM_{2.5}$-Related Mortality in China, 2013-2017, Environ. Health Perspect., 127(6), 67009, doi:10.1289/EHP4157, 2019a.

Ding, D., Xing, J., Wang, S., Chang, X. and Hao, J.: Impacts of emissions and meteorological changes on China's ozone pollution in the warm seasons of 2013 and 2017, Front. Environ. Sci. Eng., 13(5), 1–9, doi:10.1007/s11783-019-1160-1, 2019b.

Huang, J., Pan, X., Guo, X. and Li, G.: Health impact of China's Air Pollution Prevention and Control Action Plan: an analysis of national air quality monitoring and mortality data, Lancet Planet. Heal., 2(7), e313–e323, doi:10.1016/S2542-5196(18)30141-4, 2018.

Li, K., Jacob, D. J., Liao, H., Shen, L., Zhang, Q. and Bates, K. H.: Anthropogenic drivers of 2013–2017 trends in summer surface ozone in China, Proc. Natl. Acad. Sci. U. S. A., 116(2), 422–427, doi:10.1073/pnas.1812168116, 2019.

Liang, C.-K., West, J. J., Silva, R. A., Bian, H., Chin, M., Davila, Y., Dentener, F. J., Emmons, L., Flemming, J., Folberth, G., Henze, D., Im, U., Jonson, J. E., Keating, T. J., Kucsera, T., Lenzen, A., Lin, M., Lund, M. T., Pan, X., Park, R. J., Pierce, R. B., Sekiya, T., Sudo, K., and Takemura, T.: HTAP2 multi-model estimates of premature human mortality due to intercontinental transport of air pollution and emission sectors, Atmos. Chem. Phys., 18, 10497–10520, https://doi.org/10.5194/acp-18-10497-2018, 2018.

Zhang, Q., Zheng, Y., Tong, D., Shao, M., Wang, S., Zhang, Y., Xu, X., Wang, J., He, H., Liu, W., Ding, Y., Lei, Y., Li, J., Wang, Z., Zhang, X., Wang, Y., Cheng, J., Liu, Y., Shi, Q., Yan, L., Geng, G., Hong, C., Li, M., Liu, F., Zheng, B., Cao, J., Ding, A., Gao, J., Fu, Q., Huo, J., Liu, B., Liu, Z., Yang, F., He, K. and Hao, J.: Drivers of improved PM$_{2.5}$ air quality in China from 2013 to 2017, Proc. Natl. Acad. Sci. U. S. A., 116(49), 24463–24469, doi:10.1073/pnas.1907956116, 2019.

Zheng, Y., Xue, T., Zhang, Q., Geng, G., Tong, D., Li, X. and He, K.: Air quality improvements and health benefits from China's clean air action since 2013, Environ. Res. Lett., 12(114020), doi:https://doi.org/10.1088/1748-9326/aa8a32, 2017.

Zheng, B., Tong, D., Li, M., Liu, F., Hong, C., Geng, G., Li, H., Li, X., Peng, L., Qi, J., Yan, L., Zhang, Y., Zhao, H., Zheng, Y., He, K. and Zhang, Q.: Trends in China's anthropogenic emissions since 2010 as the cosnsequence of clean air actions, Atmos. Chem. Phys., 18(19), 14095–14111, doi:10.5194/acp-18-14095-2018, 2018b.

I list a few major issues which the authors may want to address-

- It might be interesting to check if there are any seasonal differences in change in PM$_{2.5}$ and ozone in China and also in downwind regions after the implementation of APPCAP.
  Response: We thank the reviewer's suggestion. We now add two plots showing seasonal air quality changes for both Population-weighted PM$_{2.5}$ and MDA8 O$_3$ in China, Japan, South Korea and U.S. (Figs. S6 & S7 in the supporting). From the plots below, we conclude that for both China and downwind regions, the surface PM$_{2.5}$ changes due to emission changes in China usually peak in the fall and winter (Fig. S7). For ozone, the emission changes from 2010 to 2013 exacerbate summer ozone pollution in China, while alleviate ozone pollution in the other three regions (Fig. S8). After 2013, the emission decreases in China exacerbate the ozone pollution for all the seasons, especially in winter. For the downwind regions, the season with peak ozone changes also varies.

  We now add the discussions of the seasonal air quality changes in the following paragraphs:
  Line 201: "The surface PM$_{2.5}$ changes in China due to emission changes usually peak in the fall and winter (Fig. S7a)."

  Line 213-215: "For ozone, the emission changes from 2010 to 2013 exacerbate summer ozone pollution in China, but alleviate ozone pollution in the other three regions (Fig. S8a). After 2013, the emission decreases in China exacerbate the ozone pollution for all the seasons, especially in winter.".

Line 235-236:" The influences are largest in spring than the other seasons for all the downwind regions (Fig. S7b-d)."

Line 239-240:" For the downwind regions, the season with peak ozone changes also varies (Fig. S8b-d)."

Fig. S7. Seasonal population-weighted PM$_{2.5}$ changes in China (a), Japan (b), South Korea (c), and U.S. (d) from 2010 to 2017 (CEDS_MEIC – CEDS_MEIC_ChinaFix).

[Figure]

Fig. S8. Same as S7 but for MDA8 O₃ changes.

[Figure]

- China has a distinct east to west gradient in air pollution exposure. Rather than speculating this east to west gradient in changes in PM₂.₅ after the APPCAP, the authors may consider representing this information (on changes in population weighted PM₂.₅, ozone and averted premature death) by provinces in China, which might be more policy relevant. The authors may also plan to inspect changes in emissions in which provinces led to maximum benefit in terms of averted death in a downwind country, if feasible. They might also plan to estimate the averted mortality/exposure in a downwind country by province. Eg- In USA the maximum benefit is expected to be realized in the western states of CA, OR and WA.

  Response: We now add a table S1-S4 in the supporting materials, listing the province-level air quality changes as well as mortality burden changes in China from 2011 to 2017. We also add the discussions of province-level results in the following texts:

  Line 203-204: "Spatially, we see that significant PM₂.₅ changes (increases before 2013 and decreases thereafter) occur in eastern China (Fig. 2), which was the focus region for China in the APPCAP (Ding et al., 2019a,b)."

Line 215-216: "The spatial pattern of ozone trends mainly featured increases in the Beijing–Tianjin–Hebei and Yangtze-River-Delta regions, and slightly decreases in the south (Fig. 3; Fig. S5; Table S2)."

The mortality burden changes in the U.S. are small (from 5 to 400 deaths/yr in Tables S4-S5), and so we will not include a state-level analysis for U.S.

Table S1. The annual PM$_{2.5}$ changes at China provinces from 2011 to 2017 (CEDS_MEIC – CEDS_MEIC_ChinaFix; unit of µg m$^{-3}$).

| Provinces | 2011 | 2012 | 2013 | 2014 | 2015 | 2016 | 2017 |
|---|---|---|---|---|---|---|---|
| Anhui | 7.32 | 5.64 | 2.19 | -4.24 | -12.14 | -17.56 | -22.84 |
| Beijing | 3.99 | 6.19 | 5.66 | 3 | -1.87 | -5.65 | -9.07 |
| Chongqing | 4.11 | 2.74 | 0.92 | -6.32 | -11.63 | -16.62 | -21.13 |
| Fujian | 1.89 | 2.66 | 0.9 | -3.5 | -4.9 | -7.44 | -9.02 |
| Gansu | 0.92 | 0.77 | 0.35 | -1.75 | -3.26 | -4.33 | -5.76 |
| Guangdong | 4.18 | 3.66 | 1.84 | -2.32 | -4.76 | -7.7 | -10.24 |
| Guangxi | 4.73 | 4.85 | 2.73 | -1.46 | -4.7 | -8.88 | -11.24 |
| Guizhou | 3.37 | 4.21 | 5.91 | 3.06 | -0.18 | -3.85 | -6.54 |
| Hainan | 1.48 | 1.77 | 2.1 | -1.07 | -1.77 | -4.24 | -6.54 |
| Hebei | 5.85 | 7.04 | 5.61 | 0.39 | -5.85 | -10.81 | -15.4 |
| Heilongjiang | 1.96 | 2.1 | 0.57 | -0.49 | -2.02 | -3.29 | -3.83 |
| Henan | 9.77 | 6.09 | 0.45 | -9.28 | -23.24 | -30.96 | -37.58 |
| Hubei | 6.82 | 4.43 | 0.5 | -9.34 | -16.31 | -20.76 | -25.82 |
| Hunan | 4.67 | 4.33 | 3.1 | -3.4 | -5.8 | -10.65 | -14.67 |
| Jiangsu | 6.54 | 4.25 | -1.26 | -4.14 | -11.08 | -15.51 | -19.87 |
| Jiangxi | 2.66 | 2.53 | 1.66 | -3.04 | -4.73 | -9.4 | -12.97 |
| Jilin | 2.65 | 3.39 | 1.73 | -1.54 | -4.13 | -5.79 | -6.48 |
| Liaoning | 2.95 | 1.71 | -0.88 | -4.06 | -6.95 | -9.31 | -10.58 |
| Nei Mongol | 1.32 | 0.14 | -0.58 | -1.12 | -1.91 | -2.77 | -3.36 |
| Ningxia Hui | 2.36 | 1.85 | 1.06 | -2.3 | -4.94 | -6.89 | -9.57 |
| Qinghai | 0.18 | 0.15 | -0.05 | -0.53 | -0.84 | -1.14 | -1.67 |
| Shaanxi | 4.08 | 4.73 | 3.25 | -4.28 | -10.6 | -13.99 | -17.33 |
| Shandong | 6.94 | 6.74 | 1.04 | -6.6 | -13.91 | -19.62 | -23.98 |
| Shanghai | 2.36 | 1.13 | -1.02 | -2.67 | -4.88 | -5.74 | -8.1 |
| Shanxi | 4.83 | 3.71 | 1.9 | -2.5 | -8.13 | -12.32 | -16.84 |
| Sichuan | 1.47 | 2.14 | 1.64 | -2.76 | -6.16 | -9.48 | -12.16 |
| Tianjin | 4.97 | 5.93 | 6.32 | 2.61 | -2.36 | -5.96 | -10.18 |
| Xinjiang Uygur | 0.53 | 1.66 | 2.11 | 1.92 | 1.65 | 1.3 | 1.06 |
| Xizang | 0.03 | 0.06 | 0.03 | -0.02 | 0.01 | -0.01 | -0.04 |
| Yunnan | 0.75 | 1.25 | 2.12 | 1.14 | -0.14 | -1.91 | -2.7 |
| Zhejiang | 2.65 | 1.79 | 0.4 | -4.49 | -7.6 | -9.21 | -12.17 |

Table S2. The same as Table S1 but for MDA8 ozone (unit of ppbv).

| Provinces | 2011 | 2012 | 2013 | 2014 | 2015 | 2016 | 2017 |
|---|---|---|---|---|---|---|---|
| Anhui | -0.34 | -0.41 | -0.08 | 0.6 | 1.34 | 1.63 | 1.83 |
| Beijing | -0.92 | -0.75 | 0.24 | 3.16 | 5.27 | 6.15 | 6.43 |
| Chongqing | 0.53 | 0.48 | 0.84 | 1.03 | 0.96 | 1.2 | 1.73 |

| | | | | | | | |
|---|---|---|---|---|---|---|---|
| Fujian | 0.26 | 0.28 | 0.26 | 0.28 | 0.28 | 0.2 | 0.2 |
| Gansu | 0.47 | 0.58 | 0.71 | 0.89 | 0.85 | 0.85 | 0.73 |
| Guangdong | 0.1 | 0.14 | 0.47 | 0.33 | 0.12 | 0.08 | -0.01 |
| Guangxi | 0.42 | 0.53 | 0.73 | 0.32 | 0.19 | -0.11 | -0.15 |
| Guizhou | 0.33 | 0.11 | 0.22 | 0.3 | 0.17 | 0.1 | 0.24 |
| Hainan | 0.34 | 0.48 | 0.35 | 0.18 | 0.15 | 0.02 | -0.13 |
| Hebei | -0.72 | -0.81 | -0.22 | 2.22 | 4.09 | 4.83 | 5.07 |
| Heilongjiang | 0.05 | -0.14 | 0 | -0.3 | 0 | 0.01 | 0.09 |
| Henan | -0.39 | 0.06 | 0.66 | 2.08 | 3.57 | 3.96 | 4.37 |
| Hubei | 0.03 | 0.32 | 1.1 | 2.09 | 2.38 | 2.49 | 2.92 |
| Hunan | 0.32 | 0.38 | 0.51 | 0.56 | 0.4 | 0.44 | 0.69 |
| Jiangsu | -0.7 | -1.16 | 0.19 | 1.3 | 2.26 | 2.65 | 3.04 |
| Jiangxi | 0.09 | 0.49 | 0.43 | 0.72 | 0.77 | 0.82 | 0.97 |
| Jilin | 0 | -0.17 | 0 | 0.34 | 0.54 | 0.54 | 0.69 |
| Liaoning | -0.07 | -0.16 | 0.1 | 0.69 | 0.97 | 1.05 | 1.27 |
| Nei Mongol | -0.17 | -0.04 | 0.07 | 0.44 | 0.54 | 0.48 | 0.5 |
| Ningxia Hui | 0.16 | 0.1 | 0.59 | 1.5 | 2.23 | 2.22 | 2.11 |
| Qinghai | 0.41 | 0.6 | 0.49 | 0.14 | -0.01 | -0.2 | -0.52 |
| Shaanxi | -0.04 | -0.15 | 0.4 | 1.84 | 2.61 | 2.95 | 3.38 |
| Shandong | -0.2 | -0.9 | 0.03 | 0.7 | 1.75 | 2.51 | 3.16 |
| Shanghai | -0.37 | -0.28 | 0.59 | 1.53 | 2.17 | 2.36 | 2.57 |
| Shanxi | -0.89 | -0.78 | -0.41 | 2.36 | 3.96 | 4.73 | 5.08 |
| Sichuan | 0.5 | 0.66 | 0.74 | 0.48 | 0.34 | 0.41 | 0.58 |
| Tianjin | -1.32 | -1.12 | 0.59 | 2.42 | 5.81 | 7.11 | 7.25 |
| Xinjiang Uygur | 0.16 | 0.32 | -0.02 | -0.08 | -0.11 | -0.06 | 0.01 |
| Xizang | 0.15 | 0.26 | 0.19 | 0 | 0.02 | -0.09 | -0.23 |
| Yunnan | 0.26 | 0.31 | 0.23 | 0.18 | 0.13 | 0.1 | 0.09 |
| Zhejiang | -0.13 | -0.22 | 0.15 | 0.93 | 1.41 | 1.58 | 1.75 |

Table S3. The annual PM$_{2.5}$ mortality burden changes at China provinces from 2011 to 2017 (CEDS_MEIC – CEDS_MEIC_ChinaFix; unit of deaths yr$^{-1}$).

| Provinces | 2011 | 2012 | 2013 | 2014 | 2015 | 2016 | 2017 |
|---|---|---|---|---|---|---|---|
| Anhui | 1261 | 1072 | 463 | -831 | -2490 | -3997 | -4944 |
| Beijing | 341 | 508 | 450 | 180 | -196 | -503 | -840 |
| Chongqing | 505 | 390 | 136 | -594 | -1217 | -2007 | -2545 |
| Fujian | 581 | 689 | 234 | -1277 | -1742 | -2761 | -3237 |
| Gansu | 371 | 370 | 94 | -587 | -1133 | -1602 | -2121 |
| Guangdong | 2652 | 2626 | 1245 | -1437 | -3529 | -5962 | -7481 |
| Guangxi | 1466 | 1741 | 830 | -756 | -1950 | -3712 | -4607 |
| Guizhou | 721 | 1055 | 1285 | 686 | 27 | -854 | -1493 |
| Hainan | 83 | 117 | 123 | -65 | -115 | -283 | -441 |
| Hebei | 1270 | 1611 | 1237 | -34 | -1397 | -2448 | -3787 |
| Heilongjiang | 738 | 914 | 297 | -17 | -540 | -1052 | -1362 |
| Henan | 1988 | 1408 | 274 | -1872 | -4703 | -6820 | -8645 |
| Hubei | 1473 | 1190 | 452 | -1836 | -3448 | -5040 | -6146 |
| Hunan | 1319 | 1365 | 979 | -1017 | -1785 | -3557 | -4791 |
| Jiangsu | 1488 | 1058 | -452 | -1431 | -3485 | -5038 | -6286 |
| Jiangxi | 719 | 788 | 532 | -976 | -1579 | -3334 | -4269 |
| Jilin | 491 | 671 | 338 | -244 | -714 | -1105 | -1314 |
| Liaoning | 568 | 263 | -246 | -798 | -1360 | -1936 | -2282 |
| Nei Mongol | 500 | 12 | -248 | -431 | -753 | -1170 | -1460 |
| Ningxia Hui | 134 | 76 | 0 | -117 | -231 | -364 | -507 |
| Qinghai | 70 | 50 | -14 | -157 | -267 | -373 | -476 |
| Shaanxi | 895 | 1161 | 584 | -731 | -1753 | -2462 | -3261 |
| Shandong | 1499 | 1575 | 159 | -1831 | -3850 | -5434 | -6833 |
| Shanghai | 251 | 134 | -148 | -417 | -753 | -960 | -1227 |
| Shanxi | 650 | 537 | 204 | -353 | -1041 | -1641 | -2351 |
| Sichuan | 973 | 1385 | 841 | -1707 | -3711 | -6001 | -7524 |
| Tianjin | 202 | 249 | 240 | 92 | -92 | -231 | -409 |
| Xinjiang Uygur | 331 | 929 | 1200 | 1170 | 1082 | 913 | 778 |
| Xizang | 8 | 11 | 9 | -4 | 11 | 5 | -3 |
| Yunnan | 421 | 741 | 1013 | 544 | -100 | -977 | -1542 |
| Zhejiang | 725 | 597 | 112 | -1549 | -2576 | -3471 | -4308 |

Table S4. The same as Table S3 but for ozone.

| Provinces | 2011 | 2012 | 2013 | 2014 | 2015 | 2016 | 2017 |
|---|---|---|---|---|---|---|---|
| Anhui | 126 | 41 | 221 | 314 | 324 | 341 | 375 |
| Beijing | -55 | -120 | -52 | 252 | 436 | 510 | 510 |
| Chongqing | 195 | 204 | 235 | 161 | 129 | 106 | 191 |
| Fujian | 197 | 194 | 58 | 50 | -2 | -92 | -36 |
| Gansu | 144 | 139 | 129 | 99 | 65 | 56 | 17 |
| Guangdong | 447 | 474 | 114 | -374 | -832 | -980 | -1038 |
| Guangxi | 256 | 331 | 316 | -9 | -145 | -336 | -393 |
| Guizhou | 240 | 348 | 459 | 276 | 144 | 13 | -6 |
| Hainan | 35 | 59 | 74 | 10 | 26 | 0 | -26 |
| Hebei | -67 | -322 | -154 | 641 | 1184 | 1473 | 1476 |
| Heilongjiang | 82 | 38 | 68 | -31 | -8 | -21 | 1 |
| Henan | 107 | 279 | 531 | 1028 | 1323 | 1461 | 1650 |
| Hubei | 332 | 283 | 412 | 338 | 327 | 269 | 351 |
| Hunan | 261 | 372 | 403 | 273 | 302 | 177 | 257 |
| Jiangsu | -99 | -153 | 352 | 726 | 861 | 934 | 904 |
| Jiangxi | 249 | 166 | 181 | 53 | 57 | -59 | -76 |
| Jilin | 68 | 41 | 49 | 46 | 40 | 20 | 42 |
| Liaoning | 64 | 16 | -8 | 114 | 114 | 109 | 164 |
| Nei Mongol | -77 | -63 | -36 | 93 | 87 | 88 | 112 |
| Ningxia Hui | 35 | 30 | 27 | 21 | 22 | 12 | -3 |
| Qinghai | 31 | 29 | 19 | 15 | 2 | -2 | -16 |
| Shaanxi | 168 | 150 | 247 | 403 | 435 | 423 | 521 |
| Shandong | 161 | -176 | 348 | 307 | 692 | 1103 | 1420 |
| Shanghai | -16 | -1 | 77 | 149 | 176 | 181 | 163 |
| Shanxi | -64 | -20 | 61 | 402 | 566 | 634 | 665 |
| Sichuan | 476 | 728 | 829 | 508 | 367 | 339 | 526 |
| Tianjin | -61 | -162 | -15 | 163 | 369 | 486 | 516 |
| Xinjiang Uygur | 22 | 38 | -41 | -42 | -47 | -30 | 8 |
| Xizang | 4 | 5 | 4 | 0 | 5 | 3 | 1 |
| Yunnan | 162 | 309 | 292 | 155 | 30 | -43 | -69 |
| Zhejiang | 56 | 72 | 239 | 291 | 365 | 342 | 330 |

- The authors may want to build few relevant emission scenarios and estimate their impact on PM$_{2.5}$/O$_3$ exposure in China and in downwind countries (eg. APPCAP is twice as effective in curbing emissions). This might inform the decision makers about the benefits of further curbing emissions in China.
  Response: We thank the reviewer's suggestion. Our study was designed to investigate the influence of realistic emission changes happening in China from 2010 to 2017, including both increasing and decreasing trend, on domestic and downwind regions' air quality. Incorporating other hypothetical emission change scenarios seems to not in the same scope of our objectives. So we decide to keep the current scenarios as they are.

Minor comments

- Line 190- please reconstruct the sentence
  Response: change as suggested. The new sentence is below:

  "At the national-scale, the Pop-weighted $PM_{2.5}$ concentration features a similar trend as the area-weighted average trend, but is notably higher (Fig. 1a), indicating that higher $PM_{2.5}$ concentrations happen in regions with higher population density."

- Line 198- please add the changes due to emission changes in the corresponding years
  Response: we now add the changes due to emission changes. Please see the new sentence below:
  "Relative to 2010, inter-annual meteorology led to annual $PM_{2.5}$ decreases as high as 4.4 µg m$^{-3}$ in 2011, and increases as high as 3.1 µg m$^{-3}$ in 2015. Meanwhile, the emission changes led to annual $PM_{2.5}$ decreases as high as 16.7 µg m$^{-3}$ in 2017, and increases as high as 4.9 µg m$^{-3}$ in 2015 (Fig. 4a)."

---

## Author Comment (AC2)

**Response to comments #2**

RC2 comments:
The work by Zhang et al. examines health impacts via air quality changes stemming from emissions changes in China from 2010-2017, expanding the role of $PM_{2.5}$ and $O_3$ and estimating the domestic vs international impacts. Overall the study is well posed. While other studies have examined this question specifically in China, here the authors focus on global-scale analysis, although in the end their findings support that a China-focused study would be sufficient, as >90% of the health impacts occur domestically. That aside, it's still likely sufficiently novel and interesting to ultimately warrant publication, however the paper itself needs some additional work in a few areas. These are described in detail in the comments below.
Response: We thank the reviewer's very positive comments of our study. We have revised the paper to take those comments into account. We provide detailed responses below (reviewers' comments in plain font, our replies in blue), and very much appreciate the reviewers' time.

Major comments:
Section 2.1: Please provide more details on which species are included in this model's estimate of $PM_{2.5}$. List primary and secondary species, both inorganic and organic. Describe how $PM_{2.5}$ itself is defined / calculated, i.e. is $H_2O$ included, at what RH, and at what temperature and pressure are all values calculated. Also, what is the height of the top of the first model layer? Are $O_3$ concentrations adjusted from this height to the surface-level (typically 2m)?
Response: We thank the reviewer's suggestion. We now add the details for the $PM_{2.5}$ calculation in line 99-107:
"Secondary organic aerosols (SOA) are derived using the two-product model approach, with laboratory derived yields for monoterpenes, isoprene, and aromatic photooxidation (Heald et al., 2008; Times et al., 2016). Recent research has suggested that anthropogenic SOA may be a dominant contributor of health impacts globally (Nault et al., 2021). As our simulations lack representation of important anthropogenic SOA precursors, such as Intermediate-Volatility Organic Compounds (IVOCs; Zhao et al., 2014; Lu et al., 2020; Pennington et al., 2021), our simulated $PM_{2.5}$ concentrations may be low biased. $PM_{2.5}$ is calculated as the sum of $SO_4+NO_3+NH_4+OC+BC+SOA+0.2*Dust+Seasalt$ (West et al., 2013; Silva et al., 2016). For dust and sea salt, only the size fractions relevant for $PM_{2.5}$ (size bins 1-3) are used. Dust in desert regions was found to be too high in the model, so global dust concentrations were multiplied by 0.2 to achieve rough consistency with the $PM_{2.5}$ concentrations estimated with Brauer et al. (2012)."

Line 110:
"The lowest modeled gridcell (~58 m above the surface) is taken to indicate ground-level concentrations."

Rereence:
        Brauer, M., Amann, M., Burnett, R. T., Cohen, A., Dentener, F., Ezzati, M., Henderson, S. B., Krzyzanowski, M., Martin, R. V, Van Dingenen, R., van Donkelaar, A. and Thurston, G. D.: Exposure assessment for estimation of the global burden of disease attributable to outdoor air pollution., Environ. Sci. Technol., 46(2), 652–60, doi:10.1021/es2025752, 2012.

Heald, C. L., Henze, D. K., Horowitz, L. W., Feddema, J., Lamar- que, J.-F., Guenther, A., Hess, P. G., Vitt, F., Seinfeld, J. H., Goldstein, A. H., and Fung, I.: Predicted change in global secondary organic aerosol concentrations in response to future climate, emissions, and land use change, J. Geophys. Res.-Atmos., 113, D05211, doi:10.1029/2007JD009092, 2008.

Silva, R. A., West, J. J., Lamarque, J.-F., Shindell, D. T., Collins, W. J., Dalsoren, S., Faluvegi, G., Folberth, G., Horowitz, L. W., Nagashima, T., Naik, V., Rumbold, S. T., Sudo, K., Takemura, T., Bergmann, D., Cameron-Smith, P., Cionni, I., Doherty, R. M., Eyring, V., Josse, B., MacKenzie, I. A., Plummer, D., Righi, M., Stevenson, D. S., Strode, S., Szopa, S., and Zengast, G.: The effect of future ambient air pollution on human premature mortality to 2100 using output from the ACCMIP model ensemble, Atmos. Chem. Phys., 16, 9847–9862, https://doi.org/10.5194/acp-16-9847-2016, 2016.

Tilmes, S., Lamarque, J. F., Emmons, L. K., Kinnison, D. E., Marsh, D., Garcia, R. R., Smith, A. K., Neely, R. R., Conley, A., Vitt, F., Val Martin, M., Tanimoto, H., Simpson, I., Blake, D. R. and Blake, N.: Representation of the Community Earth System Model (CESM1) CAM4-chem within the Chemistry-Climate Model Initiative (CCMI), Geosci. Model Dev., 9(5), 1853–1890, doi:10.5194/gmd-9-1853-2016, 2016.

West, J. J., Smith, S. J., Silva, R. A., Naik, V., Zhang, Y., Adelman, Z., Fry, M. M., Anenberg, S., Horowitz, L. W. and Lamarque, J. F.: Co-benefits of mitigating global greenhouse gas emissions for future air quality and human health, Nat. Clim. Chang., 3(10), 885–889, doi:10.1038/nclimate2009, 2013.

Somewhere this paper needs to address the significant issue of estimating PM$_{2.5}$ health impacts at such coarse spatial resolutions. Several previous studies, which are easy to find, have shown that biases can be up to 20-40% in estimates at these scales in global models.

Response: We thank the reviewer pointing this out. We now add the following sentences in line 157-163 to discuss the coarse resolution of the model we employed:

"Previous studies have shown that coarse resolution global CTMs, e.g. 1.9°×2.5°, likely generate low biases in estimating health effects, especially in urban areas (Li et al., 2016; Punger and West, 2013; Silva et al., 2013, 2016). However, less is known how these underestimates would affect the relative contributions of downwind transportation (Liang et al., 2018). Jin et al. (2019) concluded that the uncertainties in estimating the ambient PM$_{2.5}$-related mortality burden is dominated by the uncertainties in the underlying exposure-response function and less influenced by the uncertainties associated with the PM$_{2.5}$ concentration estimates."

Reference:

Jin, X., Fiore, A. M., Civerolo, K., Bi, J., Liu, Y., Van Donkelaar, A., Martin, R. V., Al-Hamdan, M., Zhang, Y., Insaf, T. Z., Kioumourtzoglou, M. A., He, M. Z. and Kinney, P. L.: Comparison of multiple PM$_{2.5}$ exposure products for estimating health benefits of emission controls over New York State, USA, Environ. Res. Lett., 14(8), 84023, doi:10.1088/1748-9326/ab2dcb, 2019.

Li, Y., Henze, D., Jack, D. and Kinney, P.: The influence of air qual- ity model resolution on health impact assessment for fine par- ticulate matter and its components, Air Qual. Atmos. Health, 9, 51–68, doi:10.1007/s11869-015-0321-z, 2016.

Liang, C.-K., West, J. J., Silva, R. A., Bian, H., Chin, M., Davila, Y., Dentener, F. J., Emmons, L., Flemming, J., Folberth, G., Henze, D., Im, U., Jonson, J. E., Keating, T. J., Kucsera, T., Lenzen, A., Lin, M., Lund, M. T., Pan, X., Park, R. J., Pierce, R. B., Sekiya, T.,

Sudo, K., and Takemura, T.: HTAP2 multi-model estimates of premature human mortality due to intercontinental transport of air pollution and emission sectors, Atmos. Chem. Phys., 18, 10497–10520, https://doi.org/10.5194/acp-18-10497-2018, 2018.

Punger, E. M. and West, J. J.: The effect of grid resolution on estimates of the burden of ozone and fine particulate matter on pre- mature mortality in the USA, Air Qual. Atmos. Health, 6, 563– 573, doi:10.1007/s11869-013-0197-8, 2013.

Silva, R. A., West, J. J., Zhang, Y., Anenberg, S. C., Lamarque, J.-F., Shindell, D. T., Collins, W. J., Dalsoren, S., Faluvegi, G., Folberth, G., Horowitz, L. W., Nagashima, T., Naik, V., Rumbold, S., Skeie, R., Sudo, K., Takemura, T., Bergmann, D., Cameron-Smith, P., Cionni, I., Doherty, R. M., Eyring, V., Josse, B., MacKenzie, I. A., Plummer, D., Righi, M., Stevenson, D. S., Strode, S., Szopa, S. and Zeng, G.: Global premature mortality due to anthropogenic outdoor air pollution and the contribution of past climate change, Environ. Res. Lett., 8(3), 034005, doi:10.1088/1748-9326/8/3/034005, 2013.

Silva, R. A., West, J. J., Lamarque, J.-F., Shindell, D. T., Collins, W. J., Dalsoren, S., Faluvegi, G., Folberth, G., Horowitz, L. W., Nagashima, T., Naik, V., Rumbold, S. T., Sudo, K., Takemura, T., Bergmann, D., Cameron-Smith, P., Cionni, I., Doherty, R. M., Eyring, V., Josse, B., MacKenzie, I. A., Plummer, D., Righi, M., Stevenson, D. S., Strode, S., Szopa, S., and Zengast, G.: The effect of future ambient air pollution on human premature mortality to 2100 using output from the ACCMIP model ensemble, Atmos. Chem. Phys., 16, 9847–9862, https://doi.org/10.5194/acp-16-9847-2016, 2016.

A recent study (Nault et al., ACP, 2021) showed that ~15% of $PM_{2.5}$ associated deaths may come from anthropogenically influenced SOA.  This component of $PM_{2.5}$ would be highly sensitive to the emissions impacted by APPCAP.  Was this accounted for in the simulated changes in total $PM_{2.5}$?

Response: That is an excellent question! The SOA module in the version of CAM_chem (CAM4, two-model product as described by Heald et al., 2008) employed in our study did not take into account of the explicit anthropogenic secondary organic aerosol (Lamarque et al., 2012; Tilmes et al., 2016). Based on the results of Nault et al. 2021, this omission could lead to underestimation of the $PM_{2.5}$-related mortality burdens at both urban and regional scales in our study. We add this discussion into line 100-102:

"The uncertainties in estimating accurate anthropogenically influenced SOA (Zheng et al., 2018; Cao et al., 2018) could make our model simulated $PM_{2.5}$ concentrations conservative (Nault et al., 2021)."

Reference:
Cao, H., Fu, T.-M., Zhang, L., Henze, D. K., Miller, C. C., Lerot, C., Abad, G. G., De Smedt, I., Zhang, Q., van Roozendael, M., Hendrick, F., Chance, K., Li, J., Zheng, J., and Zhao, Y.: Adjoint inversion of Chinese non-methane volatile organic compound emissions using space-based observations of formaldehyde and glyoxal, Atmos. Chem. Phys., 18, 15017–15046, https://doi.org/10.5194/acp-18-15017-2018, 2018.

Heald, C. L., Henze, D. K., Horowitz, L. W., Feddema, J., Lamar- que, J.-F., Guenther, A., Hess, P. G., Vitt, F., Seinfeld, J. H., Goldstein, A. H., and Fung, I.: Predicted change in global secondary organic aerosol concentrations in response to future climate, emissions, and land use change, J. Geophys. Res.-Atmos., 113, D05211, doi:10.1029/2007JD009092, 2008.

Lamarque, J.-F., Emmons, L. K., Hess, P. G., Kinnison, D. E., Tilmes, S., Vitt, F., Heald, C. L., Holland, E. a., Lauritzen, P. H., Neu, J., Orlando, J. J., Rasch, P. J. and Tyndall, G. K.: CAM-chem: description and evaluation of interactive atmospheric chemistry in the Community Earth System Model, Geosci. Model Dev., 5, 369–411, doi:10.5194/gmd-5-369-2012, 2012.

Nault, B. A., Jo, D. S., McDonald, B. C., Campuzano-Jost, P., Day, D. A., Hu, W., Schroder, J. C., Allan, J., Blake, D. R., Canagaratna, M. R., Coe, H., Coggon, M. M., DeCarlo, P. F., Diskin, G. S., Dunmore, R., Flocke, F., Fried, A., Gilman, J. B., Gkatzelis, G., Hamilton, J. F., Hanisco, T. F., Hayes, P. L., Henze, D. K., Hodzic, A., Hopkins, J., Hu, M., Huey, L. G., Jobson, B. T., Kuster, W. C., Lewis, A., Li, M., Liao, J., Nawaz, M. O., Pollack, I. B., Peischl, J., Rappenglück, B., Reeves, C. E., Richter, D., Roberts, J. M., Ryerson, T. B., Shao, M., Sommers, J. M., Walega, J., Warneke, C., Weibring, P., Wolfe, G. M., Young, D. E., Yuan, B., Zhang, Q., de Gouw, J. A., and Jimenez, J. L.: Secondary organic aerosols from anthropogenic volatile organic compounds contribute substantially to air pollution mortality, Atmos. Chem. Phys., 21, 11201–11224, https://doi.org/10.5194/acp-21-11201-2021, 2021.

Tilmes, S., Lamarque, J. F., Emmons, L. K., Kinnison, D. E., Marsh, D., Garcia, R. R., Smith, A. K., Neely, R. R., Conley, A., Vitt, F., Val Martin, M., Tanimoto, H., Simpson, I., Blake, D. R. and Blake, N.: Representation of the Community Earth System Model (CESM1) CAM4-chem within the Chemistry-Climate Model Initiative (CCMI), Geosci. Model Dev., 9(5), 1853–1890, doi:10.5194/gmd-9-1853-2016, 2016.

Other studies have shown that ammonium nitrate is a significant portion of PM$_{2.5}$ in this region. By not including nitrate, the simulated response of PM$_{2.5}$ to emissions will be rather muted. Can the authors estimate the magnitude of the uncertainty associated with this omission? I see they recognize this omission and others (lines 166-169), but it would be nice to see such approximations taken into account more quantitatively as part of their final results.

Response: We thank the reviewer for raising this issue. Lamarque et al. (2012) evaluated the performance of CAM-Chem in simulating the ammonium nitrate (CAM4, the same version used in our study) by comparing with surface observations from United States Interagency Monitoring of Protected Visual Environments (IMPROVE), and found that ammonium nitrate was fairly represented with a slightly smaller proportion for the high concentrations, possibly due to the model coarse resolution, and higher proportion for the low concentrations. However, we acknowledge that advanced treatment of nitrate in the chemical aerosol mechanism, as included in CAM5 (Tilmes et al., 2015), could potentially get accurate results with good reasons.

Reference:

Lamarque, J.-F., Emmons, L. K., Hess, P. G., Kinnison, D. E., Tilmes, S., Vitt, F., Heald, C. L., Holland, E. a., Lauritzen, P. H., Neu, J., Orlando, J. J., Rasch, P. J. and Tyndall, G. K.: CAM-chem: description and evaluation of interactive atmospheric chemistry in the Community Earth System Model, Geosci. Model Dev., 5, 369–411, doi:10.5194/gmd-5-369-2012, 2012.

Tilmes, S., Lamarque, J.-F., Emmons, L. K., Kinnison, D. E., Ma, P.-L., Liu, X., Ghan, S., Bardeen, C., Arnold, S., Deeter, M., Vitt, F., Ryerson, T., Elkins, J. W., Moore, F., Spackman, J. R., and Val Martin, M.: Description and evaluation of tropospheric chemistry and aerosols in the Community Earth System Model (CESM1.2), Geosci. Model Dev., 8, 1395–1426, https://doi.org/10.5194/gmd-8-1395-2015, 2015.

The abstract (and elsewhere) present premature mortality estimates with CI levels. These, I suspect, only reflect the uncertainty in the IERs. Given the substantial model errors and biases of up to 50%, how do these uncertainties compare to the uncertainties associated with model error? It would have been nice to see an attempt at incorporating the results of the model evaluation (section 3.1) into the subsequent analysis, rather than just touching on it in passing. They do touch on this on lines 269-275, but the text here is confusing. Why would the impact of bias be mitigated by high concentrations in China? Also, they point out here that the IER functions are non-linear. This is a reason why biases in the simulated $PM_{2.5}$ would make a difference, rather than be negligible, not this other way around.

Response: That is a good question. The IER function is super non-linear at low concentrations, but reaches linearity at higher concentrations, like in China (Burnett et al., 2014). To compare our results with other studies using higher resolution regional CTMs (Zhang et al., 2019), our study shows that from 2013 to 2017, there were 112,700 avoided deaths related to $PM_{2.5}$ reduction in China (Table 4), similar to values of 130,600 (95%I, 115,900—159,200, Table S5) estimated by Zhang et al., 2019, Our previous study has shown that in estimating ambient $PM_{2.5}$-related mortality burden, the uncertainties from concentration-response functions (such as IER) derived from epidemiology are much higher than the uncertainties from the $PM_{2.5}$ concentration estimations (Jin et al., 2019).

Reference:
    Jin, X., Fiore, A. M., Civerolo, K., Bi, J., Liu, Y., Van Donkelaar, A., Martin, R. V., Al-Hamdan, M., Zhang, Y., Insaf, T. Z., Kioumourtzoglou, M. A., He, M. Z. and Kinney, P. L.: Comparison of multiple $PM_{2.5}$ exposure products for estimating health benefits of emission controls over New York State, USA, Environ. Res. Lett., 14(8), 84023, doi:10.1088/1748-9326/ab2dcb, 2019.
    Zhang, Q., Zheng, Y., Tong, D., Shao, M., Wang, S., Zhang, Y., Xu, X., Wang, J., He, H., Liu, W., Ding, Y., Lei, Y., Li, J., Wang, Z., Zhang, X., Wang, Y., Cheng, J., Liu, Y., Shi, Q., Yan, L., Geng, G., Hong, C., Li, M., Liu, F., Zheng, B., Cao, J., Ding, A., Gao, J., Fu, Q., Huo, J., Liu, B., Liu, Z., Yang, F., He, K. and Hao, J.: Drivers of improved $PM_{2.5}$ air quality in China from 2013 to 2017, Proc. Natl. Acad. Sci. U. S. A., 116(49), 24463–24469, doi:10.1073/pnas.1907956116, 2019.

Table 2 and 3 are useful. It would also be nice to see maps of the station measurements overlaid on top of model estimated surfaces. If biases / errors are particularly large in regions most impacting export (NE), that would be useful to know, given that the main stated novelty of this work (line 87) is examining the impact of these changes on global air quality, not just domestically.

Response: Thanks for the suggestion. We now add two new plots S3-S4 in the supporting material showing the normalized mean biases (NMBs) overlaid on top of model estimated concentration (5-yr average annual $PM_{2.5}$ and MDA8 ozone from 2013 to 2017). From the plots, we see that for both $PM_{2.5}$ and ozone, the NMBs for the 5-yr (2013-2017) are both lower in the eastern coast, compared with other regions in China. We add a new sentence in line 190-191: "For both $PM_{2.5}$ and ozone, we also find that the NMBs are lower in the eastern China compared with other inland regions (Figs. S5-S6)."

Fig. S5. Evaluation of simulated annual PM$_{2.5}$ concentration against surface observations. The circles depict 249 locations of continued valid PM$_{2.5}$ observations from 2013 to 2017 (normalized mean bias(NMB), horizontal colorbar), overlaying on the 5-yr average of model simulated annual concentration (μg m$^{-3}$, vertical colorbar).

[Figure]

Fig. S6. Evaluation of simulated annual MDA8 ozone concentration against surface observations. The circles depict 700 locations of continued valid ozone observations from 2013 to 2017 (normalized mean bias(NMB), horizontal colorbar), overlaying on the 5-yr average of model simulated concentration ppbv, vertical colorbar).

[Figure]

(b)

The discussion is a bit hard to parse — it seems like the 2nd paragraph contains a lot of ideas, some which aren't well explained, and all of it mashed together in one final push of text that mixes sources of uncertainty with explanation of results and highlights of their findings. I'd recommend spending more time on this section.

Response: We thank the reviewer pointing this out. We now move the discussions of the uncertainties into 1st paragraph. In 2nd paragraph, we focus on the major findings and highlights from our study. See the updated manuscript below.

According to Fig 6, the $PM_{2.5}$ concentration impacts are significantly larger in South Korea than Japan and the US by 2017. However, in Table 4, for 2017 the mortality impacts are much bigger for Japan, and similar for South Korea and the US. Please check to make sure there wasn't a mixup. If not, please explain how this is the case (possibly given different mortality rates and populations). I see they note this point on line 282 but offer zero explanation.

Response: We appreciate the question. The higher mortality impact in Japan compared with South Korea is mainly driven by the population, with population in Japan ~2.5 times of the population in South Korea. We now add the explanation for this in line 243-246:
"Japan has a smaller change for the annual Pop-weighted $PM_{2.5}$, but much larger changes in $PM_{2.5}$-related mortality burden changes, ranging from 197 added premature deaths in 2011, and 875 avoided premature deaths in 2017, mainly caused by the much higher population in Japan than that in South Korea (https://countryeconomy.com/countries/compare/japan/south-korea?sc=XE23, last accessed Sep 3rd, 2021). "

Writing: the paper is in pretty rough shape. I doubt that many of the co-authors have contributed in detail or paid much attention to the final draft, given the prevalence of basic issues. I started making a note of corrections but tired of this after the first ten lines. Please proof-read and polish the writing throughout, it was quite distracting and at times confusing.

Response: We thank the reviewer's comment. We now ask our coauthors (Dr. Karl Seltzer) who are native speakers to polish our draft.

For example:
19: of —> of the
Response: change as suggested.

20: As a statistical aspect, it seems like human health benefits are hard to observe. Maybe rephrase?
Response: We now rephrase this sentence as the following:
"These changes have resulted in significant air quality improvements that are reflected in both surface networks and satellite observations."

20: $PM_{2.5}$ and surface —> $PM_{2.5}$, surface
Response: change as suggested

21: enough lifetime —> lifetimes
Response: change as suggested

21: which can —> to
Response: change as suggested

21: So emission —> Emission
Response: change as suggested

22: will —> will thus
Response: change as suggested

22: region air quality domestically —> domestic air quality
Response: change as suggested

22: but also —> but will also
Response: change as suggested

24: from the emission change —> from emissions changes
Response: change as suggested

24: the health —> health
Response: change as suggested

 Minor comments:
Abstract: state which model is used for this study?
Response: change as suggested. See line 23-25:
"In this study, we use a global chemistry transport model (CAM-chem) to simulate the influence of Chinese emission changes from 2010 to 2017 on both domestic and foreign air quality."

32: Not sure what is meant by "at least" in this context.  Why is this a lower bound?

Response: We use "at least" here to refer to the fact that for other years, more than 93% of PM$_{2.5}$-related mortality burdens changes happen in China, such as 94% in 2012, and 95% in 2013. We now rephrase this sentence:

"Relative to 2010, emission changes in China increased the global PM$_{2.5}$-associated premature mortality burdens through 2013, among which a majority of the changes (~93%) occurred in China. The sharp emission decreases after 2013 generated significant benefits for human-health. By 2017, emission changes in China reduced premature deaths associated with PM$_{2.5}$ by 108, 800 (92,800—124,800) deaths yr$^{-1}$ globally, relative to 2010, among which 92% were realized in China."

63: Are these numbers in response to the (arbitrary) 20% reduction in emissions used in the TF HTAP modeling tests?
Response: The reviewer is right that these numbers corresponded to the arbitrary 20% reduction employed by the TF-HTAP studies. We now make it clear in this sentence lines 56-62:
"Liang et al. (2018) used the ensemble model outputs from the Task Force on Hemispheric Transport of Air Pollution (TF HTAP, Janssens-Maenhout et al., 2015) and estimated the source-receptor relationship between air quality and avoided premature mortality from a 20% reduction in anthropogenic emissions in East Asia. They estimated that 96,600 premature mortalities from long-term PM$_{2.5}$ exposure could be avoided globally due to these emission reductions, with 6% (5,500 deaths) occurring in downwind regions. For long-term O$_3$ exposure, these emission reductions could lead to 1,400 fewer premature mortalities globally, with 15% (1,700 deaths) occurring downwind. "

147-149: GBD methods evolve annually, so I wouldn't say "latest" here.
Response: We now rephrase this sentence:
"retrieved from the Global Burden of Disease 2017 (GBD2017) study (Stanaway et al., 2018)"
Also in line 153:
"Country-age-specific baseline mortality rates (Y$_0$) in 2010 were retrieved from the GBD2017 project"

Section 2.3: Please also report: what counterfactual values were used, if any, for PM$_{2.5}$ and O$_3$? What metric was used for the O$_3$ concentration (annual average? 6 month? 1 hr or 8 hr max? etc).
Response: We thank the reviewer pointing this out. We rephrase the sentence in line 151-152:
"The *RR* for long-term O$_3$ exposure is retrieved from Turner et al., (2016), with reports a *RR* of 1.12 (95 % confidence interval (CI): 1.08, 1.16) for respiratory disease."

We also add a new sentence in line 155-157:
"The theoretical minimum risk exposure level for PM$_{2.5}$ exposure assessment is drawn from a uniform distribution with a lower bound of 5.8 μg m$^{-3}$ and an upper bound of 8.8 μg m$^{-3}$, and for O$_3$ exposure it is 26.7 ppbv."

194: What drives the isolated increase in PM$_{2.5}$ in NW China shown in Fig 2?
Response: We suspect that the PM$_{2.5}$ increases in NW China was dominated by the dust storm (Meng et al., 2019; Luo et al., 2020; Zhao et al., 2020).
We now add the explanation in line 208-209:

"There were also isolated increases in $PM_{2.5}$ in northwest China from 2010 to 2013, which were mainly caused by the dust storms (Meng et al., 2019; Luo et al., 2020; Zhao et al., 2020)"

Reference:
Meng, L., Yang, X., Zhao, T., He, Q., Lu, H., Mamtimin, A., Huo, W., Yang, F. and Liu, C.: Modeling study on three-dimensional distribution of dust aerosols during a dust storm over the Tarim Basin, Northwest China, Atmos. Res., 218(December 2018), 285–295, doi:10.1016/j.atmosres.2018.12.006, 2019.

Luo, H., Guan, Q., Pan, N., Wang, Q., Li, H., Lin, J., Tan, Z. and Shao, W.: Using composite fingerprints to quantify the potential dust source contributions in northwest China, Sci. Total Environ., 742, 140560, doi:10.1016/j.scitotenv.2020.140560, 2020.

Zhao, J., Ma, X., Wu, S. and Sha, T.: Dust emission and transport in Northwest China: WRF-Chem simulation and comparisons with multi-sensor observations, Atmos. Res., 241(March), 104978, doi:10.1016/j.atmosres.2020.104978, 2020.

244: This was confusing, until I figured out they are referring specifically to 2011. Please confirm/ clarify.
Response: We rephrase this sentence:
"The emission changes in China increased the global ozone-related mortality by 4,900 (95%CI, 3,700—5,900) premature deaths $yr^{-1}$ in 2011 (Table 5), among which 73% occurs in China (3600 premature deaths $yr^{-1}$, 95%CI: 2,700—4,300)."

247: This statement doesn't make sense. How could the changes in China alone be 43% higher than the global total change? Do they mean just the international change (excluding China) for the latter?
Response: The increased global ozone-related mortality burden (5,920 deaths $yr^{-1}$ in 2017) was lower than that in China (8,500 deaths $yr^{-1}$) was since the emission decreases in China in 2017 decreased the ozone concentration in downwind regions while increased ozone concentration in China.